# Incidence of Cardiovascular Disease in Patients with Familial Hypercholesterolemia Phenotype: Analysis of 5 Years Follow-Up of Real-World Data from More than 1.5 Million Patients

**DOI:** 10.3390/jcm8071080

**Published:** 2019-07-23

**Authors:** Luís Masana, Alberto Zamora, Núria Plana, Marc Comas-Cufí, Maria Garcia-Gil, Ruth Martí-Lluch, Anna Ponjoan, Lia Alves-Cabratosa, Roberto Elosua, Jaume Marrugat, Irene R. Dégano, Rafel Ramos

**Affiliations:** 1Lipids and Arteriosclerosis Research Unit, Sant Joan University Hospital, 43204 Reus, Catalonia, Spain; 2Internal Medicine Department, IISPV, Universitat Rovira I Virgili, CIBERDEM, 43002 Reus, Catalonia, Spain; 3Xarxa de Unitats de Lipids de Catalunya (XULA), 17300 Girona, Catalonia, Spain; 4Lipids and Arteriosclerosis Unit, Blanes Hospital, 17300 Girona, Catalonia, Spain; 5Department of Medical Sciences, School of Medicine, University of Girona, 17071 Girona, Catalonia, Spain; 6CIBER of Cardiovascular Diseases (CIBERCV), 08003 Barcelona, Catalonia, Spain; 7Laboratory of Translational Medicine (Translab), School of Medicine, University of Girona, 17071 Girona, Catalonia, Spain; 8Institut Universitari d’Investigació en Atenció Primària Jordi Gol (IDIAPJGol), 08007 Barcelona, Catalonia, Spain; 9ISV Research Group, Research Unit in Primary Care, Primary Care Services, Girona, Catalan Institute of Health (ICS), 17002 Girona, Catalonia, Spain; 10Biomedical Research Institute, Girona (IdIBGi), ICS, 17190 Girona, Catalonia, Spain; 11Registre Gironí del COR (REGICOR) Group, Cardiovascular, Epidemiology and Genetics Research Group (EGEC), Municipal Institute for Medical Research (IMIM), 08003 Barcelona, Catalonia, Spain; 12Faculty of Medicine, University of Vic-Central University of Catalonia (UVic-UCC), 08500 Vic, Catalonia, Spain

**Keywords:** Familial Hypercholesterolemia, incidence, real-world data, risk assessment, prevention, atherosclerosis, cardiovascular diseases

## Abstract

In the statin era, the incidence of atherosclerotic cardiovascular diseases (ASCVD) in patients with familial hypercholesterolemia (FH) has not been updated. We aimed to determine the incidence of ASCVD in patients with FH-phenotype (FH-P) and to compare it with that of normal low-density lipoprotein cholesterol (LDL-C) patients. We performed a retrospective cohort study using the Database of the Catalan primary care system, including ≥18-year-old patients with an LDL-C measurement. From 1,589,264 patients available before 2009, 12,823 fulfilled FH-P criteria and 514,176 patients were normolipidemic (LDL-C < 115 mg/dL). In primary prevention, patients with FH-P had incidences of ASCVD and coronary heart disease (CHD) of 14.9/1000 and 5.8/1000 person-years, respectively, compared to 7.1/1000 and 2.1/1000 person-years in the normolipidemic group. FH-P showed hazard ratio (HR) of 7.1 and 16.7 for ASCVD and CHD, respectively, in patients younger than 35 years. In secondary prevention, patients with FH-P had incidences of ASCVD and CHD of 89.7/1000 and 34.5/1000 person-years, respectively, compared to 90.9/1000 and 28.2/1000 person-years in the normolipidemic group (HR in patients younger than 35 years: 2.4 and 6.0). In the statin era, FH-P remains associated with high cardiovascular risk, compared with the normolipidemic population. This excess of risk is markedly high in young individuals.

## 1. Introduction

Familial Hypercholesterolemia (FH) is a genetic disorder characterized by high LDL cholesterol (LDL-C) plasma concentrations that leads to accelerate arteriosclerotic cardiovascular disease (ASCVD). The prevalence of the heterozygous FH form is about 1/250 individuals [1,2]. A recent re-evaluation of the WOSCOPS study showed that participants with LDL-C levels above 190 mg/dL had twice the incidence predicted by a standard risk evaluation [3]. International guidelines recommend intensive pharmacological treatment in these patients aiming to obtain an LDL-C below 100 mg/dL [4,5]. The EUROASPIRE IV study reported clinically definite FH in about 8% of patients with coronary artery disease (CAD) and 20% among those with early presentation [6]. Despite its high prevalence, FH is under-detected and undertreated. These patients are usually diagnosed around the fourth decade of life, i.e., after 40 years of hypercholesterolemia [2].

Updated data on the real incidence of cardiovascular diseases in FH populations in the statin era is limited. Several studies showed an increased prevalence of ASCVD in FH patients [2,7,8]. The SAFEHEART is the only prospective study that analyzed the incidence of ASCVD, in FH in a Mediterranean population [9], reporting a global incidence of ASCVD of about 1% per year. We aimed to assess the incidence of ASCVD and coronary heart disease (CHD) in patients with FH in primary and secondary prevention and to compare it to that of the normolidepemic population, using real world data.

## 2. Methods

### 2.1. Study Design

We carried out a retrospective cohort study.

### 2.2. Data Source

Data were obtained from the Information System for the Development of Research in Primary Care (SIDIAP). Longitudinal information of 6,177,972 patients for the period between 2005 and 2014 is structured in this database for research purposes, and includes demographic and lifestyle factors, along with diagnoses (International Classification of Diseases (ICD-10)), hospital discharge information (ICD-9), laboratory tests, and community pharmacy dispensation of prescribed medications. SIDIAP records the clinical activity of 3414 general practitioners and 853 primary care pediatricians in the 274 primary care practices of the Catalan Institute of Health, a public entity that provides healthcare services for 85% of the population in Catalonia (Spain) [10]. The quality of these data for research use has been previously documented [11,12,13]. This study complies with the Declaration of Helsinki and obtained ethics approval for observational research using SIDIAP data from a local ethics committee.

### 2.3. Study Population

The analysis included patients aged ≥18 years in January 2009, with a lipid profile determined from June 2006 to December 2008. Exclusion criteria applied to patients with a history of hypothyroidism, nephrotic syndrome, or basal triglyceride values ≥400 mg/dL.

### 2.4. Follow up and Outcomes

Follow-up for ASCVD and CHD events was from January 2009 to December 2014. Patients were censored at the earliest date of the diagnosis of interest, transfer, or study ending (31 December 2014). Time to first event was considered for all analyses.

During follow-up we analyzed the incidence of fatal and non-fatal CHD (myocardial infarction or angina), and the incidence of ASCVD (CHD, ischemic stroke, and peripheral artery disease), based on the ICD-9 and ICD-10 codes obtained from primary care registers and hospital discharge reports (see in Appendix A).

### 2.5. Variables and Definitions

Participants were defined as receiving lipid-lowering therapy (LLT) if their records showed at least two purchases of statin or ezetimibe within 6 months before their LDL-C measurement. We classified patients’ exposure to statins or ezetimibe according to the cholesterol reduction capacity of the drug, as follows: low, <30%; moderate, 30–50%; high, 50–60%; and very high, >60% [14]. Adherence to treatment was calculated according to the Medical Possession Ratio (MPR): Number of days of statin supply during 6 consecutive months, divided by 183 days.

We categorized hypercholesterolemia status using each patient’s untreated LDL-C record closest and prior to January 2009. Untreated LDL-C was defined as no record of LLT purchases during at least 6 months before the LDL-C test. Ten multiple imputation by chained equations were used to replace the missing values of pre-treated LDL-C of individuals that consistently had no record of untreated LDL-C. The imputation of pre-treated cholesterol levels for participants on medication at baseline has been shown to yield estimates consistent with reports from randomized clinical trials [15]. The imputation models included age, sex, dose and type of LLT, and treatment adherence (MPR).

We defined FH according to validated age-adjusted LDL-C thresholds previously delimited in the Spanish population. The SIDIAP database does not contain information on familial history of premature CVD, presence of cholesterol deposits, or genetic tests. We referred to individuals with FH as FH phenotype (FH-P) patients. In adults, heterozygous FH-P was defined as untreated if LDL-C >230 mg/dL for 18- to 29-years-old; >238 mg/dL for 30- to 39-years-old, >260 mg/dL for 40- to 49-years-old, and >255 mg/dL for participants older than 49 years [16]. For epidemiological comparisons, individuals with basal LDL-C < 115 mg/dL were considered normolipidemic, as stated in the guidelines of European Cardiology and of Atherosclerosis Societies [5].

The following baseline variables were obtained from the SIDIAP database: age, sex, systolic and diastolic blood pressure (SBP and DBP, respectively), body mass index (BMI), and laboratory tests: Total cholesterol (TC), LDL-cholesterol (LDL-C), HDL-cholesterol (HDL-C), triglycerides (TG), glucose, glycosylated hemoglobin (HBA1c), and creatinine. We also recorded the presence of cardiovascular risk factors (CVRF): Diabetes mellitus (DM), hypertension, hypercholesterolemia, obesity, and smoking.

Previous history of ASCVD was defined as composite of fatal and non-fatal coronary heart disease (myocardial infarction or angina), ischemic stroke, and peripheral artery disease. We defined some clinical conditions to characterize the studied population: a person with polyvascular ASCVD was any case with at least two vascular territories affected; recent CHD disease was a coronary event in the last year; progressive CHD disease indicated at least two coronary events in the last year.

### 2.6. Statistical Analyses

Dichotomous variables were expressed as percentages and compared using the 2-sample test for equality of proportions with Yates’ continuity correction; continuous variables were reported as mean and standard deviation (SD) and compared using a *t*-test. The final significance regarding variables with missing values was obtained using Rubin rules.

Raw and adjusted incidences were estimated using a Poisson model. A likelihood ratio test was used to examine if incidences between men and women were significantly different. The incidence rates (IRs) in FH-P and normolipidemic populations were adjusted by age and sex. Cumulative hazard functions were built from monthly Kaplan–Meier estimates. Hazard ratios were obtained from a Cox proportional hazards regression model and adjusted for age, sex, presence of hypertension, DM, smoking and presence of obesity. We also performed the main analysis stratified by statin use. In addition, we compared the complete-case results with those including multiple imputation as a sensitivity analysis. Statistical analyses were carried out using R-software 3.4.3 [17].

## 3. Results

Among 1,589,264 patients that fulfilled selection criteria, 514,176 subjects had LDL-C ≤ 115 mg/dL and were selected as the reference normolipidemic group. The criteria for FH-P applied to 12,823 patients, 2202 of whom had previous ASCVD. Incident ASCVD occurred in 1495 patients with FH-P, 750 in primary prevention and 745 in secondary prevention; and in 28,061 patients from the normolipidemic group, 15,980 in primary prevention and 13,081 in secondary (Figure 1). Table 1 shows the baseline characteristics of the study population.

In primary prevention, the IRs of ASCVD were 14.9 per 1000 person-years in the group with FH-P, and 7.1 per 1000 person-years in the normolipidemic group, HR (95% CI): 1.45 (1.34–1.58), Table 2.

In secondary prevention, the IRs of ASCVD were 89.8 per 1000 person-years in the group with FH-P and 91.0 per 1000 person-years in the reference group, HR (95%CI): 1.11 (1.01–1.21), Table 3.

The HRs for ASCVD were clearly higher in younger than in older patients, to the utmost in the <35-year-old group (HR: 7.13 (95% CI 3.23–15.73) (Table 2). This excess of risk weakened with age, and it almost disappeared at 75 years old in primary prevention (Table 2), and around 45 years of age in secondary prevention (Table 3). Appendix A compares the characteristics of participants with and without complete pre-treated LDL data in both FH and control groups. Only individuals who consistently received LLT and had no record of previous untreated LDL-C were imputed. As expected, the main findings indicate that 100% of patients with missing values were on LLT; whereas in the complete-case population, such was the case for only 46.8% of participants with FH-P, and 2.3% of participants with normal LDL-C values. Accordingly, participants with imputed pre-treatment LDL-C were older, included higher percentage of men, in the control group, and had higher prevalence of cardiovascular risk factors (diabetes, hypertension, obesity) but lower mean LDL-C values. The complete-case analysis showed slightly increased HRs compared with the analysis with multiple imputations (Appendix A).

The effect of FH-P upon the ASCVD incidence by LLT use showed a marked excess of risk in patients without treatment but only a slight increase in patients with statin treatment (Figure 2). The excess of risk also attenuated with age in both groups. As in the general analysis, the effect on secondary prevention was less marked. The models of the group receiving LLT were adjusted by cardiovascular risk factors, and additionally by statin potency and adherence to treatment.

In primary prevention, each additional risk factor almost doubled the previous risk of ASCVD. In the absence of other risk factors, the presence of FH-P increased the risk of ASCVD 3.75 times compared to the normolipidemic population. The presence of diabetes increased the incidence rate of ASCVD more than the presence of any other two CVRF (Figure 3). In secondary prevention, the FH-P groups with higher risk of a second ASCVD event were: Concomitant presence of ≥2 CVRF, diabetes, recent ASCVD, progressive ASCVD, or polyvascular ASCVD (Figure 4).

Similar relative results were found for CHD. The IRs in the group with FH-P and in the normolipidemic group were 5.82 and 2.07 per 1000 person-years, respectively, HR: 1.95 (95% CI 1.65–2.28) (Appendix A). Interaction with age was even higher than that for ASCVD; HR (95% CI) in patients <35-year-old: 16.70 (6.66–41.85).

In secondary prevention, the incidence rates for CHD were 34.5 and 28.2 per 1000 person-years, respectively (HR: 1.29 (95% CI 1.08–1.53) (Appendix A).

The HRs for CHD in secondary prevention were also visibly higher in younger than in older. i.e., HRs (95% CI) for CHD in the <35-year-old group was 5.98 (0.81–44.06). This excess of risk disappeared around 55 years (Appendix A).

Groups at high risk for CHD were those with >2 additional CVRF and diabetic patients in primary prevention (Appendix A). In secondary prevention individuals at extreme risk of CHD were also those with concomitant presence of >2 CVRF, diabetes, recent ASCVD, progressive ASCVD, or polyvascular ASCVD (Appendix A).

## 4. Discussion

We observed that patients with FH-P had notably increased risks for ASCVD compared to the normolipidemic population. The increase of risk was markedly high in young individuals, younger than 35 years, reaching seven-fold for premature ASCVD in primary prevention and two-fold in secondary prevention. This excess of risk was mitigated from 75 years old in primary prevention and around 45 years old in secondary prevention. An incidence compatible with extreme risk was observed in patients with FH-P in secondary prevention and additional CVRF or with recent or progressive or polyvascular ASCVD. Similar results were found for CHD. The excess of CHD risk, in patients younger than 35 years, reached 16-fold in primary prevention and 6-fold in secondary prevention.

Our results showed a high incidence of ASCVD and CHD in FH-P patients in primary prevention (IR: 14.9 and 5.8 per 1000 person-years, respectively); and in secondary prevention (IR: 89.7 and 34.5 per 1000 person-years, respectively). The SAFEHEART study provided data on incidence of cardiovascular disease in persons with FH in a Spanish population. In a cohort of 2404 genetically diagnosed patients with FH (307 of which were in secondary prevention), the overall incidence of ASCVD for primary and secondary prevention was 5.6 and 43 per 1000 person-years, respectively [9]. Several reasons may explain these differences: the population with FH in our study represents the real-world situation of these patients as opposed to those genetically diagnosed, who can start lipid-lowering treatment earlier, and are managed in Lipid Clinics. The mean age in the SAFEHEART study was 45 years, compared to 62 years in our study, and we obtained a lower proportion of patients on optimal lipid-lowering therapy [9].

The impact of FH-P on the incidence of ASCVD and CHD was more evident in primary prevention and in younger people in our study. Despite the moderate absolute numbers, the menace of ASCVD events at young ages recommends preventative actions in these patients, including early therapy. In advanced ages, this excess of risk was tamed, and it almost disappeared after 75 years of age in primary prevention, and around 45 years of age in secondary prevention. The global older mean age of the population in our study could explain the lower effect of FH-P on ASCVD compared to previous reports in genetically identified individuals who were younger than those included in our analysis; whereas the effect in the younger subgroup was similar. Our results in primary prevention are in agreement with previous reports, in a Norwegian population with FH [18] and recently in the Simon Broome Register [19] and in primary care subjects in UK [20]. There is no previous data on the excess of ASCVD incidence in persons with FH-P in secondary prevention.

The incidence of ASCVD and CHD was higher in men than in women, however the HRs were similar in both sexes, therefore, no grounds should ASCVD risk be minimized in women with FH-P [21].

The presence of additional risk factors, observed in about 80% of our population, highlights those patients at particular high risk. One recent meta-analysis showed that smoking, hypertension, and diabetes accounted for more than one quarter of the ASCVD risk in individuals with FH, and LDL-C levels >154 mg/dL accounted for 33% of such ASCVD risk [22]. Another study showed that the prevalence of ASCVD in patients with FH and diabetes doubled that of patients with FH and no diabetes [23]. A recent report that also included patients with high clinical probability of FH showed the significant impact of classical cardiovascular risk factors on recurrent coronary revascularization in these patients [24]. Statin therapy has changed the natural history of FH, as recently shown in an analysis of data from the Simon Broome Register [19]. In this report, the CHD standardized mortality of patients with FH was significantly higher before 1992, the pre-statin era. A recent study from the Dyslipidaemia Registry of Spanish Arteriosclerosis Society showed that the presence of additional CV risk factors accounted for incremental risk even in patients with FH on long-term statin therapy; and that treatment with statins for at least 5 years initiated before age 30 decreases ASCVD incidence by half [25]. Our data contributes to characterize the risk in individuals with FH-P. Even in patients with FH-P and no additional risk factors, particularly in the younger ones, the risk was remarkably increased compared to those with normal LDL-C levels. Moreover, the association of at least one additional risk factor doubled the overall risk. Our results advocate a multifactorial approach on cardiovascular risk in patients with FH-P.

The impact of FH-P in secondary prevention was less marked probably because the ASCVD has its own pathophysiological evolution, dependent on other additional factors, such as thrombotic or inflammatory conditions. However, FH-P has a significant impact on younger people. The excess risk was of 139% for ASCVD and 498% for a CHD event in those younger than 35 years in secondary prevention. These data are consistent with that in the SPUM-ACS study [26]. In our study, Individuals with recent or progressive CHD presented the highest incidence of recurrent ASCVD, which highlights the need for specific therapy strategies in this population, including intensive lipid-lowering therapy. This is also consistent with a previous study in which the main manifestation of ASCVD in patients with probable FH was premature multivessel CAD [24].

Recent results from trials with long follow-up on PCSK9 inhibitors (iPCSK9) in patients with ASCVD have shown a significant 15% relative risk reduction for major cardiovascular events, especially in patients with recent or progressive CHD or with involvement of several vascular territories [27,28]. Even though these trials were not carried out specifically in a cohort of patients with FH [29], these groups match those detected in our study as FH-P of extreme risk of ASCVD. We observed that 50% and 96% of persons with FH-P in primary and secondary prevention, respectively, would be candidates for iPCSK9 treatment after optimizing lipid-lowering treatment and adherence [30].

The results from the stratified analysis by LLT use are of particular interest. These results agree with previous reports that observed association of prolonged lipid-lowering treatment with a huge reduction in ASCVD in patients with FH [24]. In patients who were not receiving LLT, the excess of risk was critical, particularly in primary prevention. Of note, most treated patients in our cohort had been at least 3 years on statins, but the majority of them had probably been on statins for a very long time (because we do not have information about treatment before 2006). These results support the idea of early and sustained treatment with LLT as the key point in the management of FH-P. This observation was supported by the complete-case analysis, in which the proportion of patients without lipid lowering therapy was lower than in the imputed dataset (46.8% in participants with FH-P and 2.3% in control group), and the group with FH-P showed a higher excess of risk compared with the imputed analysis. These differences arise the need to include this part of the population in order to avoid a selection bias.

Regarding treatment recommendations, lipid-lowering therapy of high intensity should be considered in patients with FH-P and diabetes in primary prevention, and in those with two or more additional risk factors. Patients with FH-P and at least one previous ASCVD event, especially with progressive or recent CHD, or those with ASCVD involving several vascular territories, should be considered for very high-intensity lipid-lowering therapy.

We acknowledge that the assignment of FH on clinical bases, mainly LDL-C, is a limitation in our study. We included all individuals who had an LDL-C test in the last 4 years; hence, we cannot discard some selection bias in the age and sex composition of the population that would tend to overestimate the FH-P prevalence. However, the age- and sex-standardized prevalence of FH-P was 1/186 (0.54%), which coincides with recent published data [2,7,8]. The diagnostic method, based on the HF phenotype, could also contribute to overestimate the real prevalence, especially due to inclusion of persons with polygenic familial hypercholesterolemia. However, a concept of familial hypercholesterolemic syndrome (which includes heterozygous familial hypercholesterolemia, homozygous familial hypercholesterolemia, polygenic familial hypercholesterolemia, and familial hypercholesterolemia combined with hypertriglyceridemia) has been recently defined on the basis that all these types of hypercholesterolemia present a clinically relevant excess of risk [31,32]. Furthermore, such possible overestimation of the prevalence would not significantly affect the estimates of the effect of FH-P on the ASCVD incidence. We also acknowledge that the diagnostic method may increase the number of false negative rates, in coherence with studies that show relatively high rates of genetically diagnosed heterogenic FH in patients with lower LDL-C levels. This is important, as data on family history was lacking in our study.

Our study population should also be considered: they were persons attending the primary care services, mostly consulting for reasons other than high cholesterol levels, introducing a bias in terms of co-morbidities. Strengths of our study include the use of real-world data, warranting representativeness of this population of patients with FH-P. The sample size is another important strength of our work, which renders robustness of the data.

In conclusion, in the statin era, patients with FH-P presented a high incidence of ASCVD and CHD, especially those with additional risk factors or those with previous ASCVD. This excess of risk was extremely high in young individuals, especially in those younger than 35 years. This excess of risk weakened with age, and it almost disappeared at 75 years old in primary prevention, and around 45 years of age in secondary prevention. By sex, the risk of ASCVD and CHD was higher in men than in women, but women with FH-P compared to the normolipidemic population presented similar excess of risk than men.

## Figures and Tables

**Figure 1 jcm-08-01080-f001:**
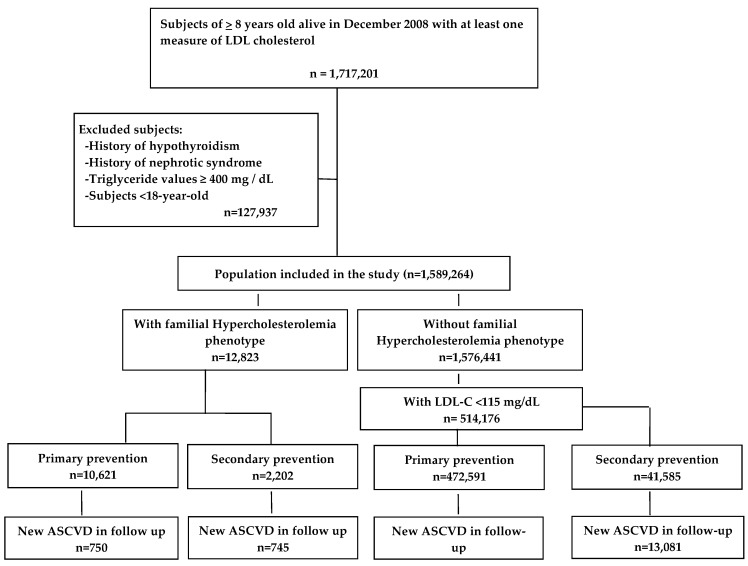
Study flowchart. ASCVD indicates Atherosclerotic Cardiovascular Disease. LDL: Low-density lipoprotein; LDL-C: Low-density lipoprotein cholesterol; ASCVD: Atherosclerotic cardiovascular diseases.

**Figure 2 jcm-08-01080-f002:**
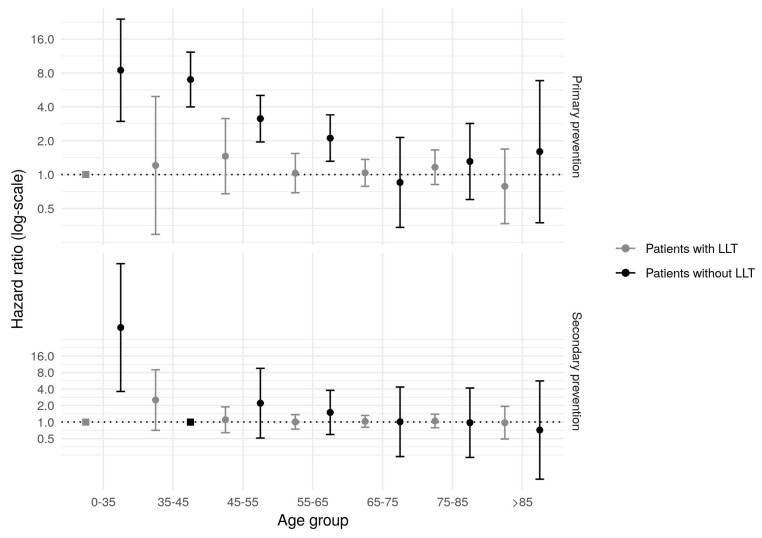
Effect of the presence of Familial Hypercholesterolemia phenotype by statin use, in primary and secondary prevention. LLT: Lipid-lowering therapy. All the models in primary prevention have been adjusted for age, sex, presence of hypertension, diabetes mellitus, smoking, and presence of obesity. Adjustment for disease characteristics (progressive or recent) has been added in the secondary prevention models. The models of patients with lipid- lowering therapy have been further adjusted by statin potency and adherence to treatment (medical possession ratio).

**Figure 3 jcm-08-01080-f003:**
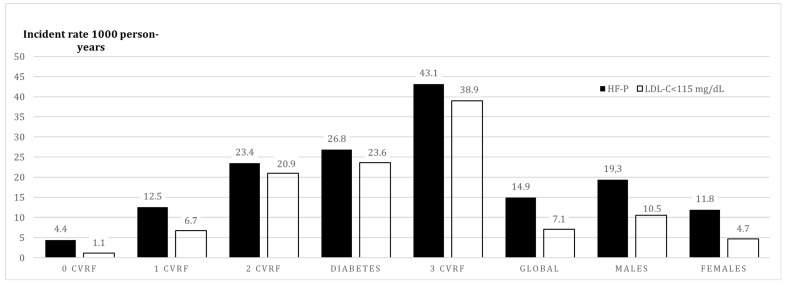
Incidence rate (per 1000 person-years) of ASCVD in primary prevention in FH-P vs. normolipidemic population based on the presence of number of cardiovascular risk factors. ASCVD: Atherosclerotic Cardiovascular Disease; CVRF: Cardiovascular Risk Factor; FH-P: Familial Hypercholesterolemia Phenotype; LDL-C: Low-Density Lipoprotein Cholesterol.

**Figure 4 jcm-08-01080-f004:**
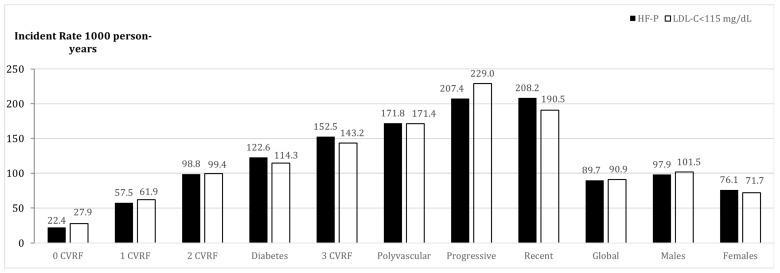
Incidence rate (per 1000 person-years) of ASCVD in secondary prevention in FH-P vs. normolipidemic population based on the presence of number of cardiovascular risk factors and clinical associated characteristics. ASCVD: Atherosclerotic Cardiovascular Disease; CVRF: Cardiovascular Risk Factor; HF-P: Familial Hypercholesterolemia; CLDL: Cholesterol-LDL.

**Table 1 jcm-08-01080-t001:** Baseline characteristics of the study population according to the presence of familial hypercholesterolemia phenotype (FH-P) and previous arteriosclerotic cardiovascular disease (ASCVD).

	Normal LDL-C < 115 mg/dL	FH-P		Normal LDL-C < 115 mg/dL	FH-P	
Variable	Without ASCVD	Without ASCVD	*P* Value	With ASCVD	With ASCVD	*P* Value
*N*	472,591	10,621		41,585	2202	
Age, mean (SD)	48.2 (19.3)	60.5 (13.6)	<0.01	73.2 (12.1)	67.7 (10.7)	<0.01
Male, sex, %	40.9%	40.9%	0.98	64.5%	63.3%	0.34
Diabetes, mellitus, %	14.2%	21.7%	<0.01	44%	36.7%	<0.01
Hypertension, %	33.6%	60.7%	<0.01	91.8%	93.3%	0.03
Current, smoker, %	30.3%	32.3	<0.01	43.3%	49.1	<0.01
BMI, mean (SD)	27.5 (5.6)	28.9 (4.7)	<0.01	28.8 (4.9)	29.2 (4.5)	<0.01
Receiving lipid therapy, %	4.1%	87.1%	<0.01	36.3%	97.4%	<0.01
Personal history of CAD, %	-	-	-	36.3%	46.9%	<0.01
Personal history of PAD, %	-	-	-	16.7%	15.6%	0.27
Personal history of Stroke, %	-	-	-	30.4%	22.8%	<0.01
TC, mg/dL, mean (SD)	169.9 (26.1)	257.9 (59.5)	<0.01	160.8 (29.5)	223.9 (5)	<0.01
LDL-C, mg/dL, mean (SD)	91.8 (17.9)	177.6 (55.5)	<0.01	85.4 (20.4)	147.1 (46.0)	<0.01
Untreated LDL-C, mg/dL, mean (SD)	92.3 (16.6)	285.2 (34.7)	<0.01	93.08 (16.7)	287.5 (3)	<0.01
HDL-C, mg/dL, mean (SD)	56.1 (16.3)	56.3 (13.8)	0.23	49.3 (15.2)	51.6 (13.0)	<0.01
Non-HDL-C, mg/dL, mean (SD)	113.7 (24.2)	201.7 (59.3)	<0.01	111.4 (26.7)	172.3 (51.3)	<0.01
TG, mg/dL, mean (SD)	111.2 (74.1)	149.1 (88)	<0.01	127.3 (81.8)	150.6 (83.1)	<0.01
Creatinine, mean (SD)	0.85 (0.3)	0.89 (0.2)	<0.01	1.08 (0.6)	1.03 (0.4)	<0.01
HBA1c, mean (SD)	5.7 (1.5)	6.02 (1.5)	<0.01	6.2 (1.4)	6.2 (1.6)	0.55

LDL-C: Low-density lipoprotein cholesterol; TC: Total Cholesterol; HDL-C: High-density lipoprotein cholesterol; TG: triglycerides; HbA1C: Glycosylated haemoglobin; BMI: Body Mass Index; CAD: Coronary Artery Disease; PAD: Peripheral Arterial Disease.

**Table 2 jcm-08-01080-t002:** Incidence rate (1000 person-years) and hazard ratio of atherosclerotic cardiovascular disease by age in primary prevention.

	Normal LDL-C Population (<115 mg/dL)			Familial Hypercholesterolemia Phenotype			
Age Groups	Number (*n*)	Events (*n*)	IR (1000 p/y) 95%CI	Number (*n*)	Events (*n*)	IR (1000 p/y) IC 95	HR IC 95
All	472,591	15,980	7.1 (7.0–7.2)	10,621	749	14.9 (13.7–16.2)	1.45 (1.33–1.58)
0–35	141,920	143	0.2 (0.2–0.2)	551	6	2.1 (0.9–5.2)	7.13 (2.85–17.84)
35–45	91,274	483	1.1 (1.0–1.2)	833	30	7.3 (4.7–11.2)	3.78 (2.42–5.91)
45–55	70,331	1256	3.6 (3.4–3.8)	1652	76	9.4 (7.1–12.5)	1.86 (1.41–2.46)
55–65	56,835	2635	9.6 (9.3–10.0)	3241	198	12.7 (10.5–15.3)	1.43 (1.19–1.72)
65–75	50,897	4237	18.0 (17.4–18.6)	2713	232	18.3 (15.6–21.4)	1.16 (0.98–1.36)
75–85	44,604	5232	27.8 (27.0–28.6)	1441	181	28.4 (23.1–34.8)	1.13 (0.92–1.39)
>85	16,730	1995	36.1 (34.4–37.8)	189	27	37.8 (23.6–60.6)	1.09 (0.68–1.75)

LDL-C: Low-density Lipoprotein Cholesterol; IR (1000 p/y): Incident Rate 1000 persons-years; CI: Confidence Interval; HR: Hazard Ratio. The models have been adjusted for age, sex, presence of hypertension, diabetes mellitus, smoking and presence of obesity.

**Table 3 jcm-08-01080-t003:** Incidence rate (1000 person/years) and hazard ratio of atherosclerotic cardiovascular disease by age in secondary prevention.

	Normal LDL-C Population (<115 mg/dL)			Familial Hypercholesterolemia Phenotype			
Age Groups	Number (*n*)	Events (*n*)	IR (1000 p/y) 95%CI	Number (*n*)	Events (*n*)	IR (1000 p/y) IC 95	HR IC 95
All	41,586	13,081	91.0 (89.2–92.8)	2202	745	89.8 (82.1–98.1)	1.10 (1.01–1.21)
0–35	262	22	18.2 (11.7–28.4)	8	2	74.6 (14.4–385.9)	2.36 (0.37–14.95)
35–45	698	134	43.9 (36.5–52.8)	36	14	98.6 (51.3–189.4)	1.92 (0.98–3.76)
45–55	2329	631	66.4 (60.8–72.4)	214	68	79.3 (59.8–105.2)	1.14 (0.86–1.51)
55–65	5898	1772	75.7 (71.8–79.8)	581	188	81.8 (67.5–99.1)	1.12 (0.93–1.35)
65–75	10,340	3480	90.1 (86.9–93.5)	718	244	88.7 (75.8–103.8)	1.07 (0.91–1.27)
75–85	14,965	5128	103.9 (100.7–107.2)	544	193	101.6 (84.9–121.5)	1.06 (0.88–1.27)
>85	7094	1912	102.8 (97.9–107.9)	100	35	111.4 (68.1–182.1)	1.13 (0.67–1.92)

LDL-C: Low-density Lipoprotein Cholesterol; IR 1000 p/y: Incident Rate 1000 persons-years; CI: Confidence Interval; HR: Hazard Ratio. The models have been adjusted for age, sex, presence of hypertension, diabetes mellitus, smoking, the presence of obesity, and disease characteristics (progressive or recent).

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
