# Peer review of "Incidence of Cardiovascular Disease in Patients with Familial Hypercholesterolemia Phenotype: Analysis of 5 Years Follow-Up of Real-World Data from More than 1.5 Million Patients"

_jcm, 2019, doi:10.3390/jcm8071080_

Round 1
Reviewer 1 Report
In the current study the authors investigated retrospectively the incidence of ASCVD in a large cohort of patients with clinically diagnosed FH. The data on the features of ASCVD in FH patients in the statin era is clinically relevant. The main conclusion of the study is that FH remains associated with high CV risk, and that this excess in risk is markedly high in young individuals with FH. This was similarly demonstrated previously in several study cohorts; Nevertheless, the study is novel in the comparison to normolipidemic population (and not only non-FH individuals) and the evaluation of both primary and secondary prevention at baseline. In addition, the sample size is considerable.
I have several comments for the authors to consider:
1. Introduction line 48: people with LDL-C>190 mg/dl do not necessarily have FH, and this cutoff may detect mostly polygenic hypercholesterolemia. Therefore, it seems the phrase “who are very likely to suffer from FH” should be omitted.
2. Introduction: authors should define abbreviations when first mentioned in text: CHD. In addition, lines 56+58 ACVD probably means ASCVD (ACVD also appears at at tables).
3. Methods, line 85: How was angina defined? Was there a definite diagnosis of ASCVD in patient with angina? In addition, authors should elaborate on how peripheral artery disease as defined (included low ABI? aneurysm? )
4. Using imputation to define untreated LDL-C may lead to bias. Authors may consider to perform a sensitivity analysis without missing LDL-C data.
5. The rate of clinically diagnosed FH in the current study was very high 1:124 individuals screened. This is although the authors used age-dependent very high LDL-C cutoffs ( 230-260 mg/dl according to age, significantly higher than >190 mg/dl), and excluded secondary causes of severe hypercholesterolemia. Accordingly, the authors should explain/discuss this surprising high rate of extreme hypercholesterolemia. May this be associated with the use of imputation for untreated LDL-C levels?
6. Line 103, the term heterozygous should not be used, as this was not a genetic diagnosis.
7. Surprisingly, FH patients had higher rated of obesity (increased BMI) and diabetic/pre-diabetic measures than non-FH patients. This is in contrast to what was seen in “classic” FH patients in the past. Is it because the increase in metabolic syndrome in developed countries, or because part of the patients are not “genetically” heterozygous FH patients? The authors may discuss these points.
8. Discussion: The authors may further refer to previous data on manifestations of ASCVD in FH patients, as was investigated in several cohorts (for example, similar inclusion/exclusion criteria used in - Clinical Features and Gaps in the Management of Probable Familial Hypercholesterolemia and Cardiovascular Disease (Circ J. 2017 Dec 25;82(1):218-223).
9. Limitations: It should be further emphasized that using very-high LDL-C cutoffs may decrease false positive (FP) diagnosis of FH, but increases significantly false negative (FN) rates, as in genetic studies we see relatively high rates of genetically diagnosed HeFH patients with lower LDL-C levels. This is important, as data on family history was lacking.
10. Lines 261-265 (Regarding treatment recommendations…..) : this conclusions are not based on the data presented in the results, and therefore seems inappropriate as final conclusions (may appear in discussion).
Author Response
Title: Incidence of cardiovascular disease in patients with familial hypercholesterolemia phenotype: analysis of 5 years follow-up of real-world data from more than 1.5 million patients
Manuscript ID: jcm-537696
Response to the Editor and the Reviewers
Dear Editor,
We appreciate your comments and encouragement to reply to the questions raised by the reviewers’ comments, which have contributed to improving the clarity of our message and presentation of our findings.
During this revision process we have detected an error in the calculus of the confidence intervals for the incidence estimations and hazard ratios, which has been corrected in the current version. This correction has no relevant implications.
We have prepared a point-by-point response:
First Reviewer:
Comments and Suggestions for Authors
In the current study the authors investigated retrospectively the incidence of ASCVD in a large cohort of patients with clinically diagnosed FH. The data on the features of ASCVD in FH patients in the statin era is clinically relevant. The main conclusion of the study is that FH remains associated with high CV risk, and that this excess in risk is markedly high in young individuals with FH. This was similarly demonstrated previously in several study cohorts; Nevertheless, the study is novel in the comparison to normolipidemic population (and not only non-FH individuals) and the evaluation of both primary and secondary prevention at baseline. In addition, the sample size is considerable.
The authors appreciate this assessment of our work.
I have several comments for the authors to consider:
1. Introduction line 48: people with LDL-C>190 mg/dl do not necessarily have FH, and this cutoff may detect mostly polygenic hypercholesterolemia. Therefore, it seems the phrase “who are very likely to suffer from FH” should be omitted.
We appreciate this comment. We have omitted the phrase from the introduction.
2. Introduction: authors should define abbreviations when first mentioned in text: CHD. In addition, lines 56+58 ACVD probably means ASCVD (ACVD also appears at at tables).
Thank you for this remark. We have defined the abbreviation (CHD) and corrected the typo ACVD, using ASCVD instead, in the text and in the tables.
3. Methods, line 85: How was angina defined? Was there a definite diagnosis of ASCVD in patient with angina? In addition, authors should elaborate on how peripheral artery disease as defined (included low ABI? aneurysm? )
Thank you for this question. Angina and peripheral artery disease definitions were based on the ICD-9 and ICD-10 codes obtained from primary care registers and hospital discharge reports. The quality of these codes for cardiovascular research use has been previously documented (1). ASCVD definition included (myocardial infarction, angina, ischemic stroke, and peripheral artery disease), thus having a diagnosis of angina was enough to be considered in the group with ASCVD. The used codes for peripheral artery disease did not include aneurysms because we were interested in arteriosclerotic cardiovascular diseases. The definition of ASCVD slightly differs between guidelines. The ACC/AHA defines this as acute myocardial infarction, unstable angina, previous myocardial infarction, stable angina, coronary or other revascularization, ischaemic stroke or transient ischaemic attack, and atherosclerotic peripheral arterial disease. Our ASCVD definition coincides with this of the ACC/AHA. By the other hand, the ESC/EAS include all those included in the ACC/AHA guidelines, but also include any pre-clinical evidence for atherosclerotic disease based on any imaging technique (2).
To clarify these definitions, we have detailed the ICD-9 and ICD-10 codes in the supplemental material.
1. Ramos, R.; Ballo, E.; Marrugat, J.; Elosua, R.; Sala, J.; Grau, M.; Vila, J.; Bolíbar, B.; García-Gil, M.; Martí, R; et al. Validity for use in research on vascular diseases of the SIDIAP (Information System for the Development of Research in Primary Care): the EMMA study. Rev. Esp. Cardiol. 2012, 65:29–37.
2. Ray KK, Kastelein JJ, Boekholdt SM, Nicholls SJ, Khaw KT, Ballantyne CM, Catapano AL, Reiner Ž, Lüscher TF. The ACC/AHA 2013 guideline on the treatment of blood cholesterol to reduce atherosclerotic cardiovascular disease risk in adults: the good the bad and the uncertain: a comparison with ESC/EAS guidelines for the management of dyslipidaemias 2011. Eur Heart J. 2014;35:960-8.
4. Using imputation to define untreated LDL-C may lead to bias. Authors may consider to perform a sensitivity analysis without missing LDL-C data.
Thank you for this suggestion. The main purpose of the multiple imputation process in our study was to minimize the potential selection bias that might have occurred if we had not imputed the variables with missing information. Individuals with no missing pre-treated LDL-C values are mainly those without lipid lowering therapy and are a minority among FH-P individuals. Hence, we believe that data from the imputed dataset is more representative of the FH-P population, who was our main interest.
We used 10 multiple imputations by chained equations to replace missing pre-treated LDL-C values (3) in treated individuals with no record of untreated LDL-C (i.e., they consistently purchased medication). The imputation of pre-treated cholesterol levels for participants on medication at baseline has been shown to yield estimates consistent with reports of randomized clinical trials (3).
However, we agree with the reviewer about performing a sensitivity analysis as the best option to clarify the impact of multiple imputation on the results of the study.
Table S1 compares the characteristics of participants with and without complete pre-treated LDL data in both FH and control groups. Only individuals who consistently received LLT and had no record of previous untreated LDL-C were imputed. As expected, the main findings indicate that 100% of patients with missing values were on LLT; whereas in the complete-case population, such was the case for only 46.8% of participants with FH-P, and 2.3% of participants with normal LDL-C values. Accordingly, participants with imputed pre-treatment LDL-C were older, included higher percentage of men, in the control group, and had higher prevalence of cardiovascular risk factors (diabetes, hypertension, obesity) but lower mean LDL-C values. These differences arise the need to include this part of the population in order to avoid a selection bias.
The complete-case analysis showed a slightly higher relative effect of FH-P on the incidence of ASCVD. We think that this occurred mainly because the complete-case analysis mostly included individuals without lipid-lowering therapy, Tables S2 and S3.
Table S1. Comparison of Participants Characteristics between population with and without complete pre-treated LDL data in both Familial Hypercholesterolemia Phenotype and Normal LDL-C population groups.
Normal LDL-C population (<115 mg/dL) | Familial Hypercholesterolemia Phenotype | ||||||
Variable | Complete-case | Imputed-case | p-value | Complete-case | Imputed-case | p-value | |
N | 490,959 | 23,218 | 2,652 | 10,171 | |||
Age, mean (SD) | 49.3 (19.9) | 71.2 (10.9) | <0.0001< span=""> | 51.7 (15.5) | 64.4 (11.5) | <0.0001< span=""> | |
Sex (male), % | 42.2% | 56.7% | <0.0001< span=""> | 44.3% | 44.9% | 0.6126 | |
Receiving lipid lowering therapy, % | 2.3% | 100.0% | - | 46.8% | 100.0% | - | |
Diabetes mellitus, % | 15.1% | 50.9% | <0.0001< span=""> | 11.5% | 27.7% | <0.0001< span=""> | |
Hypertension, % | 35.9% | 89.7% | <0.0001< span=""> | 39.6% | 73.3% | <0.0001< span=""> | |
Current smoker, % | 31.1% | 37.5% | <0.0001< span=""> | 41.7% | 33.6% | <0.0001< span=""> | |
Body mass index, mean (SD) | 27.6 (5.6) | 29.5 (4.8) | <0.0001< span=""> | 28.1 (4.8) | 29.2 (4.6) | <0.0001< span=""> | |
Obesity, % | 21.8% | 40.5% | <0.0001< span=""> | 23.4% | 34.5% | <0.0001< span=""> | |
TC, mg/dL, mean (SD) | 169.5 (26.0) | 162.7 (34.4) | <0.0001< span=""> | 314.6 (73.3) | 236.0 (43.2) | <0.0001< span=""> | |
LDL-C, mg/dL, mean (SD) | 91.8 (16.9) | 81.9 (25.3) | <0.0001< span=""> | 236.3 (65.8) | 155.8 (37.2) | <0.0001< span=""> | |
HDL-C, mg/dL, mean (SD) | 55.8 (16.4) | 51.1 (15.5) | <0.0001< span=""> | 54.8 (14.4) | 55.8 (13.7) | 0.0049 | |
TG, mg/dL, mean (SD) | 111.1 (73.4) | 143.1 (95.2) | <0.0001< span=""> | 162.3 (106.5) | 146.0 (81.2) | <0.0001< span=""> | |
Creatinine, mg/dL, mean (SD) | 0.9 (0.4) | 1.0 (0.6) | <0.0001< span=""> | 0.9 (0.3) | 0.9 (0.3) | 0.0041 | |
HBA1c, mean (SD) | 5.8 (1.5) | 6.4 (1.5) | <0.0001< span=""> | 6.0 (2.0) | 6.1 (1.5) | 0.5010 | |
SD: standard deviation; BMI: Body Mass Index; TC: Total Cholesterol; LDL-C: Low-density lipoprotein cholesterol; HDL-C: High-density lipoprotein cholesterol; TG: triglycerides; HbA1C: Glycosylated haemoglobin.
Table S2 Incidence rate (1000 person-years) and hazard ratio of atherosclerotic cardiovascular disease by age in primary prevention. Complete-case analysis.
Normal LDL-C population (<115 mg/dL) | Familial Hypercholesterolemia Phenotype | ||||||
Age Groups | Number (n) | Events (n) | IR (1000 p-y) 95%CI | Number (n) | Events (n) | IR (1000 p-y) 95%CI | HR 95%CI |
All | 453,183 | 13,912 | 6.4 (6.3-6.5) | 2,253 | 121 | 11.3 (9.3-13.6) | 2.23 (1.85-2.69) |
0-35 | 141,845 | 142 | 0.2 (0.2-0.2) | 433 | 5 | 2.3 (0.9-5.8) | 9.20 (3.59-23.53) |
35-45 | 91,022 | 476 | 1.1 (1.0-1.2) | 473 | 17 | 7.3 (4.4-12.0) | 4.69 (2.81-7.83) |
45-55 | 69,489 | 1,205 | 3.5 (3.3-3.7) | 531 | 29 | 11.4 (7.7-16.7) | 2.59 (1.75-3.82) |
55-65 | 54,442 | 2,433 | 9.3 (8.9-9.7) | 586 | 39 | 14.1 (10.1-19.6) | 1.83 (1.31-2.55) |
65-75 | 46,720 | 3,779 | 17.5 (16.9-18.1) | 282 | 20 | 15.6 (9.8-24.9) | 1.13 (0.71-1.80) |
75-85 | 40,631 | 4,689 | 27.4 (26.6-28.2) | 158 | 19 | 29.2 (18.2-46.8) | 1.36 (0.85-2.19) |
>85 | 15,785 | 1,850 | 35.6 (33.9-37.3) | 26 | 3 | 31.7 (8.4-120.2) | 0.98 (0.26-3.69) |
LDL-C: Low-density Lipoprotein Cholesterol; IR 1000 p/y: Incident Rate persons-years; CI: Confidence Interval; HR: Hazard Ratio.
Table S3. Incidence rate (1000 person-years) and hazard ratio of atherosclerotic cardiovascular disease by age in secondary prevention. Complete-case analysis.
Normal LDL-C population (<115 mg/dL) | Familial Hypercholesterolemia Phenotype | ||||||
Age Groups | Number (n) | Events (n) | IR (1000 p-y) 95%CI | Number (n) | Events (n) | IR (1000 p-y) 95%CI | HR 95%CI |
All | 26,468 | 7,506 | 82.9 (80.9-84.9) | 125 | 41 | 90.9 (65.9-125.3) | 1.37 (0.99-1.89) |
0-35 | 253 | 20 | 16.8 (10.6-26.7) | 3 | 2 | 342.9 (80.1-1466.9) | 25.48 (4.47-145.33) |
35-45 | 616 | 111 | 40.8 (33.6-49.6) | 6 | 3 | 133.5 (35.8-497.4) | 2.85 (0.74-11.01) |
45-55 | 1,800 | 464 | 62.7 (57.0-69.0) | 23 | 8 | 90.9 (43.6-189.2) | 1.26 (0.60-2.66) |
55-65 | 4,228 | 1,202 | 71.1 (67.0-75.4) | 45 | 16 | 100.7 (59.8-169.7) | 1.61 (0.94-2.75) |
65-75 | 7,169 | 2,259 | 83.6 (80.0-87.3) | 38 | 12 | 80.5 (44.2-146.6) | 1.07 (0.59-1.95) |
75-85 | 10,961 | 3,544 | 98.0 (94.6-101.4) | 36 | 12 | 109.3 (58.1-205.6) | 1.33 (0.71-2.50) |
>85 | 5,998 | 1,552 | 100.2 (95.1-105.6) | 13 | 5 | 122.4 (44.9-333.4) | 1.45 (0.53-3.96) |
LDL-C: Low-density Lipoprotein Cholesterol; IR 1000 p/y: Incident Rate persons-years; CI: Confidence Interval; HR: Hazard Ratio.
Following the reviewer’s suggestion, we have added the tables of the complete-case analysis in the supplementary material and commented about this in the Methods, Results and Discussion sections as follows:
Methods (statistical analysis):
“In addition, we compared the complete-case results with those including multiple imputation as a sensitivity analysis.”
Results:
“Table S1 compares the characteristics of participants with and without complete pre-treated LDL data in both FH and control groups. Only individuals who consistently received LLT and had no record of previous untreated LDL-C were imputed. As expected, the main findings indicate that 100% of patients with missing values were on LLT; whereas in the complete-case population, such was the case for only 46.8% of participants with FH-P, and 2.3% of participants with normal LDL-C values. Accordingly, participants with imputed pre-treatment LDL-C were older, included higher percentage of men, in the control group, and had higher prevalence of cardiovascular risk factors (diabetes, hypertension, obesity) but lower mean LDL-C values. The complete-case analysis showed slightly increased HRs compared with the analysis with multiple imputations (Tables S2 and S3)”
Discussion:
“…This observation was supported by the complete-case analysis, in which the proportion of patients without lipid lowering therapy was lower than in the imputed dataset (46.8% in participants with FH-P and 2.3% in control group), and the group with FH-P showed a higher excess of risk compared with the imputed analysis. These differences arise the need to include this part of the population in order to avoid a selection bias.”
3. Jorgensen NW et al, Sibley CT, McClelland RL. Using imputed pre-treatment cholesterol in a propensity score model to reduce confounding by indication: results from the multi-ethnic study of atherosclerosis. BMC Med Res Methodol. 2013;13:81
5. The rate of clinically diagnosed FH in the current study was very high 1:124 individuals screened. This is although the authors used age-dependent very high LDL-C cutoffs ( 230-260 mg/dl according to age, significantly higher than >190 mg/dl), and excluded secondary causes of severe hypercholesterolemia. Accordingly, the authors should explain/discuss this surprising high rate of extreme hypercholesterolemia. May this be associated with the use of imputation for untreated LDL-C levels?
We appreciate this comment. In a previous study using the same data source, the observed global FH-P prevalence in 2014 was 0.58% (95%CI: 0.58-0.60), or 1/172 individuals, and the age- and sex-standardized prevalence of FH-P was slightly lower: 0.52% (95%CI: 0.51-0.53) (1/192). That study included all individuals who had an LDL-C test within the 9 years previous to the study entry, which represented 68.2% of the subjects older than 45 years (4). In this current study we have included individuals who had an LDL-C test within the 4 years previous to their entry, hence, we cannot discard some selection bias in the population that could mainly affect the age and sex distribution. As the reviewer rightly states, the observed FH-P prevalence in our study was 1/124 (0.8%), but the age- and sex-standardized prevalence of FH-P was considerably lower: 1/186 (0.54%) which coincides with recent published data. In any case, we believe that, a possible overestimation of the prevalence would not significantly affect the estimates of the effect of FH-P on the ASCVD incidence.
Following the reviewer’s observation, we have added a comment in the limitations section.
“We included all individuals who had an LDL-C test in the last 4 years; hence, we cannot discard some selection bias in the age and sex composition of the population that would tend to overestimate the FH-P prevalence. However, the age- and sex-standardized prevalence of FH-P was 1/186 (0.54%), which coincides with recent published data [2,7,8]...”
“…. Furthermore, such possible overestimation of the prevalence would not significantly affect the estimates of the effect of FH-P on the ASCVD incidence.”
4. Zamora A, Masana L, Comas-Cufí M, Vila À, Plana N, García-Gil M, Alves-Cabratosa L, Marrugat J, Roman I, Ramos R; XULA and ISV-Girona groups. Familial hypercholesterolemia in a European Mediterranean population-Prevalence and clinical data from 2.5 million primary care patients. J Clin Lipidol. 2017;11:1013-1022.
6. Line 103, the term heterozygous should not be used, as this was not a genetic diagnosis.
Thank you for this remark. We have removed the term.
7. Surprisingly, FH patients had higher rated of obesity (increased BMI) and diabetic/pre-diabetic measures than non-FH patients. This is in contrast to what was seen in “classic” FH patients in the past. Is it because the increase in metabolic syndrome in developed countries, or because part of the patients are not “genetically” heterozygous FH patients? The authors may discuss these points.
We see the reviewer’s point. In our sample of a primary care population, the prevalence of CVRF (obesity, diabetes) differed from previous studies, likely because we identified patients with FHP on the basis of LDL-C measurements from a primary care population. Conversely, previous studies included genetic identification and referral from specialized clinical settings, primary care, or including information from their FH-affected relatives. This probably contributes to explain the differences in the characteristics of the two study populations that could be explained by the older age of the FH phenotype population in our study (mean age of FH-P individuals in our study was 60.5 years in the group without previous ASCVD and 67.7 in the group with previous ASCVD).
However, we agree with the reviewer that some patients classified as FH phenotype population might suffer polygenic familial hypercholesterolemia. However, a concept of familial hypercholesterolemic syndrome (which includes heterozygous familial hypercholesterolemia, homozygous familial hypercholesterolemia, polygenic familial hypercholesterolemia, and familial hypercholesterolemia combined with hypertriglyceridemia) has been recently defined on the basis that all these types of hypercholesterolemia present a clinically relevant excess of risk (5).
To address the reviewer’s concern, we have added a comment in the limitations section:
“The diagnostic method, based on the HF phenotype, could also contribute to overestimate the real prevalence, especially due to inclusion of persons with polygenic familial hypercholesterolemia. However, a concept of familial hypercholesterolemic syndrome (which includes heterozygous familial hypercholesterolemia, homozygous familial hypercholesterolemia, polygenic familial hypercholesterolemia, and familial hypercholesterolemia combined with hypertriglyceridemia) has been recently defined on the basis that all these types of hypercholesterolemia present a clinically relevant excess of risk [31,32].”
31. Masana L, Ibarretxe D, Rodríguez-Borjabad C, Plana N, Valdivielso P, Pedro-Botet J, Civeira F, López-Miranda J, Guijarro C, el al. Toward a new clinical classification of patients with familial hypercholesterolemia: One perspective from Spain. Atherosclerosis. 2019;287:89-92.
32 Khera AV, Chaffin M, Aragam KG, Haas ME, Roselli C, Choi SH, Natarajan P, Lander ES, Lubitz SA, Ellinor PT, Kathiresan S. Genome-wide polygenic scores for common diseases identify individuals with risk equivalent to monogenic mutations. Nat Genet. 2018;50:1219-1224.
8. Discussion: The authors may further refer to previous data on manifestations of ASCVD in FH patients, as was investigated in several cohorts (for example, similar inclusion/exclusion criteria used in - Clinical Features and Gaps in the Management of Probable Familial Hypercholesterolemia and Cardiovascular Disease (Circ J. 2017 Dec 25;82(1):218-223).
We appreciate this comment. We have included a comment in the Discussion section, including the above-mentioned reference:
“A recent report that also included patients with high clinical probability of FH showed the significant impact of classical cardiovascular risk factors on recurrent coronary revascularization in these patients [24].”
And
“This is also consistent with a previous study in which premature multivessel CAD was the main manifestation of ASCVD in patients with probable FH [24].”
24. Zafrir B, Jubran A, Lavie G, Halon DA, Flugelman MY, Shapira C. Clinical Features and Gaps in the Management of Probable Familial Hypercholesterolemia and Cardiovascular Disease. Circ J. 2017;82:218-223.
9. Limitations: It should be further emphasized that using very-high LDL-C cutoffs may decrease false positive (FP) diagnosis of FH, but increases significantly false negative (FN) rates, as in genetic studies we see relatively high rates of genetically diagnosed HeFH patients with lower LDL-C levels. This is important, as data on family history was lacking.
We agree with the reviewer’s suggestion. We have added the following comment in the limitations section:
“We also acknowledge that the diagnostic method may increase the number of false negative rates, in coherence with studies that show relatively high rates of genetically diagnosed heterogenic FH in patients with lower LDL-C levels. This is important, as data on family history was lacking in our study.”
10. Lines 261-265 (Regarding treatment recommendations…..) : this conclusions are not based on the data presented in the results, and therefore seems inappropriate as final conclusions (may appear in discussion).
We appreciate this comment. We have moved them to the Discussion section.

Reviewer 2 Report
The manuscript is interesting, timely and relevant in that it provides real world information in a large population of phenotypic FH patients. The main point of the study is that patients who have phenotypic FH are at increased risk of CV events. Those at the highest risk were younger patients who likely have true monogenic FH and not a polygenic cause for their high LDL cholesterol and those FH patients with additional CV risk factors.
However one main issue with the manuscript is that it does not provide information on statin use and CV outcomes. In the methods section the patients’ use of statins and ezetimibe along with adherence were recorded. However none of this information is presented in the results section. The authors need to present this information. They then need to discuss how statin use has impacted CV outcomes, both in the FH cohort and in the control group. For example 13% of patients with phenotypic FH and without ASCVD were not on a statin. Was there a difference in CV outcomes among those FH patients who are on a statin vs. those on no statin therapy? The authors should also provide information on statin intensity and adherence and how that relates to outcomes. Another limitation to the study is that the control group was not age matched. It is somewhat surprising that the IR of ASCVD in the control group without ASCVD was as high as 7.1 per 1000 person years and that the IR risk of ASCVD in the FH cohort was only 2 fold higher. I would think that given the older age of the FH cohort along with their other CV risk factors (which were much more prevalent compared to the control group) that for primary prevention the risk would have been higher.
Minor points:
1. In figure 1 in the top box it should be subjects > 18 years old not 8.
Author Response
Title: Incidence of cardiovascular disease in patients with familial hypercholesterolemia phenotype: analysis of 5 years follow-up of real-world data from more than 1.5 million patients
Manuscript ID: jcm-537696
Response to the Editor and the Reviewers
Dear Editor,
We appreciate your comments and encouragement to reply to the questions raised by the reviewers’ comments, which have contributed to improving the clarity of our message and presentation of our findings.
During this revision process we have detected an error in the calculus of the confidence intervals for the incidence estimations and hazard ratios, which has been corrected in the current version. This correction has no relevant implications.
We have prepared a point-by-point response:
Second reviewer:
The manuscript is interesting, timely and relevant in that it provides real world information in a large population of phenotypic FH patients. The main point of the study is that patients who have phenotypic FH are at increased risk of CV events. Those at the highest risk were younger patients who likely have true monogenic FH and not a polygenic cause for their high LDL cholesterol and those FH patients with additional CV risk factors.
The authors appreciate this assessment of our work. The comments of the reviewer have substantially improved the quality of our work.
However one main issue with the manuscript is that it does not provide information on statin use and CV outcomes. In the methods section the patients’ use of statins and ezetimibe along with adherence were recorded. However none of this information is presented in the results section. The authors need to present this information. They then need to discuss how statin use has impacted CV outcomes, both in the FH cohort and in the control group. For example 13% of patients with phenotypic FH and without ASCVD were not on a statin. Was there a difference in CV outcomes among those FH patients who are on a statin vs. those on no statin therapy? The authors should also provide information on statin intensity and adherence and how that relates to outcomes.
We thank the reviewer for this comment, this is a very relevant question. We agree with this suggestion and thus have added the description of the lipid lowering therapy potency and patient treatment adherence to Table 1. We have also classified the patients’ exposure to statins or ezetimibe considering the cholesterol reduction capacity of the drug, as follows: low,<30%; and="" very="">60% [14].
We have further carried out the analyses suggested by the reviewer examining the effect of FH-P on the incidence of ASCVD by statin use. The results of such analysis show that the presence of FH-P associates with a remarkable excess of risk in patients without treatment but only with a slight risk increase in patients with statin treatment (Figure 2). The models of the group receiving lipid-lowering medications were adjusted for cardiovascular risk factors, and by statin potency and adherence. This excess of risk also attenuates with age in both groups (treated and non-treated). As in the general analysis, the effect in secondary prevention was less marked.
Figure 2. Hazard ratios of the presence of Familial Hypercholesterolemia phenotype on atherosclerotic cardiovascular disease by statin use, in primary and secondary prevention.
LLT: Lipid Lowering therapy
These results agree with previous reports on the association of prolonged lipid-lowering treatment with a huge reduction in ASCVD in patients with FH (1). Note that most treated patients in our cohort had been at least 3 years on statins, but probably most of them had been on statins for a very long time (because we do not have information about treatment before 2006).
We have included this analysis in the manuscript, with the corresponding explanations in the text.
Methods:
“We classified patients’ exposure to statins or ezetimibe according to the cholesterol reduction capacity of the drug, as follows: low,<30%; and="" very="">60% [14]”
And
“We also performed the main analysis stratified by statin use.”
Results:
“The effect of FH-P upon the ASCVD incidence by LLT use showed a marked excess of risk in patients without treatment but only a slight increase in patients with statin treatment (Figure 2). The excess of risk also attenuated with age in both groups. As in the general analysis, the effect on secondary prevention was less marked. The models of the group receiving LLT were adjusted by cardiovascular risk factors, and additionally by statin potency and adherence to treatment.”
Discussion:
“The results from the stratified analysis by LLT use are of particular interest. These results agree with previous reports that observed association of prolonged lipid-lowering treatment with a huge reduction in ASCVD in patients with FH [24]. In patients who were not receiving LLT, the excess of risk was critical, particularly in primary prevention. Of note, most treated patients in our cohort had been at least 3 years on statins, but the majority of them had probably been on statins for a very long time (because we do not have information about treatment before 2006). These results support the idea of early and sustained treatment with LLT as the key point in the management of FH-P.”
14. Stone NJ, Robinson JG, Lichtenstein AH, Bairey Merz CN, Blum CB, Eckel RH, Goldberg AC, Gordon D, Levy D, Lloyd-Jones DM, et al. 2013 ACC/AHA guideline on the treatment of blood cholesterol to reduce atherosclerotic cardiovascular risk in adults: a report of the American College of Cardiology/American Heart Association Task Force on Practice Guidelines. J Am Coll Cardiol 2014; 63: 2889–2934.
23. Perez-Calahorra S, Laclaustra M, Marco-Benedí V, Lamiquiz-Moneo I, Pedro-Botet J, Plana N, Sanchez-Hernandez RM, Amor AJ, Almagro F, Fuentes F, Suarez-Tembra M, Civeira F; Dyslipemia Registry of Spanish Arteriosclerosis Society. Effect of lipid-lowering treatment in cardiovascular disease prevalence in familial hypercholesterolemia. Atherosclerosis. 2019 May;284:245-252
Another limitation to the study is that the control group was not age matched. It is somewhat surprising that the IR of ASCVD in the control group without ASCVD was as high as 7.1 per 1000 person years and that the IR risk of ASCVD in the FH cohort was only 2 fold higher. I would think that given the older age of the FH cohort along with their other CV risk factors (which were much more prevalent compared to the control group) that for primary prevention the risk would have been higher.
Many thanks for this comment. The reviewer is right, the control group was not age matched and the incidences were crude, but the hazard ratios were age adjusted. In response to the reviewer’s concern, we have included the information of the adjusting variables in the table’s footnotes.
This comment is very interesting. Previous studies included genetic identification and referrals from specialized clinical settings, primary care, and evaluation of their FH-affected relatives. This probably explains the differences between the characteristics of our study population and other reports: the mean age of FH-P individuals in our study was 60.5 in the group without previous ASCVD, and 67.7 in group with previous ASCVD, while in most of the previous studies the mean age was below 50 years. In our study, the global incidence of ASCVD in FH-P individuals was only slightly higher than in the control group, mainly because of the effect in older patients. Our results show that the excess of risk in young individuals, is marked [e.g., HR (95%CI) in<35 years: 7.13 (2.85-17.84) in 35-45 years 3.78 (2.42-5.91)]. A similar interaction the hazard ratios of FH with age has been observed previously (17).
We have included this comment in the Discussion section:
“The global older mean age of the population in our study could explain the lower effect of FH-P on ASCVD compared to previous reports in genetically identified individuals who were younger than those included in our analysis; whereas the effect in the younger subgroup was similar.”
A second reason would relate with the previous reviewer’s comment. Long term LLT, which is present in most patients with FH-P, may have reduced the lipid burden, which in turn, likely reduced the incidence of ASCVD in this group. In this regard, we have added a comment in the text, as stated in the previous response.
17.Mundal, L.J.; Igland, J.; Veierød, M.B.; Holven, K.B.; Ose, L.; Selmer, R.M.; Wisloff, T.; Kristiansen, I.S.; Tell, G.S,; Leren, T.P.; Retterstøl, K. Impact of age on excess risk of coronary heart disease in patients with familial hypercholesterolaemia. Heart 2018, 104: 1600–1607.
Minor points:
1. In figure 1 in the top box it should be subjects > 18 years old not 8.
We appreciate this remark. Please note that we included persons<8 years old at the beginning of the population selection, and excluded those <18 years on a posterior step (see criteria of excluded patients in the flowchart).

Round 2
Reviewer 1 Report
The authors have made an effort to address all comments that were raised.
Reviewer 2 Report
I would like to thank the authors for their well constructed revisions to their manuscript.